Patterns of evolution of MHC class II genes of crows (Corvus) suggest trans-species polymorphism

Eimes John A. 1 johneimes12@gmail.com
Townsend Andrea K. 2
Sepil Irem 3
Nishiumi Isao 4
Satta Yoko 1
1 Department of Evolutionary Studies of Biosystems, Graduate University for Advanced Studies (SOKENDAI) , Hayama , Japan
2 Department of Biology, Hamilton College , Clinton, NY , USA
3 Department of Zoology, University of Oxford , Oxford , UK
4 Department of Zoology, National Museum of Nature and Science , Tsukuba , Japan
Edwards Scott
Electronic publication date: 2015 Mar 19
Publication date: 2015
Volume: 3
Electronic Location ID: e853
Received 2014 Nov 20; Accepted 2015 Mar 4
Copyright: © 2015 Eimes et al.
Copyright year: 2015
Copyright holder: Eimes et al.
License: This is an open access article distributed under the terms of the Creative Commons Attribution License, which permits unrestricted use, distribution, reproduction and adaptation in any medium and for any purpose provided that it is properly attributed. For attribution, the original author(s), title, publication source (PeerJ) and either DOI or URL of the article must be cited.
License URL: https://creativecommons.org/licenses/by/4.0/

Keywords: Crow, MHC, Trans-species polymorphism, Species divergence, Convergent evolution, Corvus, Balancing selection, Supertyping

Funding: Ministry of Education, Culture, Sports, Science and Technology of Japan 22133007 This work was supported by Grant-in-Aid for Scientific Research on Innovative Areas from the Ministry of Education, Culture, Sports, Science and Technology of Japan (22133007) to Yoko Satta. The funders had no role in study design, data collection and analysis, decision to publish, or preparation of the manuscript.

==============================
A distinguishing characteristic of genes that code for the major histocompatibility complex (MHC) is that alleles often share more similarity between, rather than within species. There are two likely mechanisms that can explain this pattern: convergent evolution and trans-species polymorphism (TSP), in which ancient allelic lineages are maintained by balancing selection and retained by descendant species. Distinguishing between these two mechanisms has major implications in how we view adaptation of immune genes. In this study we analyzed exon 2 of the MHC class IIB in three passerine bird species in the genus Corvus: jungle crows (Corvus macrorhynchos japonensis) American crows (C. brachyrhynchos) and carrion crows (C. corone orientalis). Carrion crows and American crows are recently diverged, but allopatric, sister species, whereas carrion crows and jungle crows are more distantly related but sympatric species, and possibly share pathogens linked to MHC IIB polymorphisms. These patterns of evolutionary divergence and current geographic ranges enabled us to test for trans-species polymorphism and convergent evolution of the MHC IIB in crows. Phylogenetic reconstructions of MHC IIB sequences revealed several well supported interspecific clusters containing all three species, and there was no biased clustering of variants among the sympatric carrion crows and jungle crows. The topologies of phylogenetic trees constructed from putatively selected sites were remarkably different than those constructed from putatively neutral sites. In addition, trees constructed using non-synonymous substitutions from a continuous fragment of exon 2 had more, and generally more inclusive, supported interspecific MHC IIB variant clusters than those constructed from the same fragment using synonymous substitutions. These phylogenetic patterns suggest that recombination, especially gene conversion, has partially erased the signal of allelic ancestry in these species. While clustering of positively selected amino acids by supertyping revealed a single supertype shared by only jungle and carrion crows, a pattern consistent with convergence, the overall phylogenetic patterns we observed suggest that TSP, rather than convergence, explains the interspecific allelic similarity of MHC IIB genes in these species of crows.

Introduction

The major histocompatibility complex (MHC) is an unusual example of a functional gene complex that exhibits high levels of polymorphism. The MHC is a multigene cluster that encodes molecules that bind and present peptides to T-cells in vertebrates, initiating a cascade of immunological responses to pathogens (Janeway et al., 2005). The MHC is the most polymorphic coding gene family in vertebrate genomes (Klein, 1986) and the maintenance of this polymorphism is usually attributed to pathogen-mediated balancing selection (Sommer, 2005), although sexual selection, as well as molecular mechanisms such as recombination and gene conversion, may play important roles (Andersson & Mikko, 1995; Martinsohn et al., 1999; Zelano & Edwards, 2002).

One distinguishing characteristic of MHC genes is that alleles often share more similarity between rather than among species. Two mechanisms have been proposed to explain this “trans-specific” similarity of MHC alleles. The first explanation, called “trans-species polymorphism” (TSP), proposes that orthologous MHC allelic lineages are maintained by balancing selection, often over macro-evolutionary time-scales, and persist through speciation events (Klein et al., 1998). Strong evidence for TSP in MHC IIB genes has been found in a wide range of taxa, particularly those of mammals and fish (Klein et al., 1993; Graser et al., 1996; Garrigan & Hedrick, 2003; Lenz et al., 2013).

An alternative explanation for trans-specific MHC allele clustering is convergent or parallel evolution. The term molecular convergent evolution has been used to describe similar, but distinct phenomena (e.g., convergence between non-orthologous genes or within orthologues only) (Zhang & Kumar, 1997; Yeager & Hughes, 1999; Kriener et al., 2000), and debate exists for distinguishing molecular parallel evolution from molecular convergent evolution as well (Arendt & Reznick, 2008; Pearce, 2011). For the purpose of this study, we define convergence as any interspecific allele similarity that arises from adaptation to similar selective pressures (Gustafsson & Andersson, 1994; Hughes, 1999).

A common method used for identifying TSP, and in some cases for distinguishing TSP from convergence, is the comparison of codons thought to be under selection, such as those found in the MHC peptide binding region (PBR), with synonymous substitutions in surrounding codons or flanking introns. Alleles that are similar by decent should retain a signature of ancestry at selectively neutral sites. On the other hand, convergence is indicated if alleles are more similar at putatively selected sites than at neutral sites (Klein et al., 1998). Specifically, convergence and TSP can be tested by constructing different phylogenetic trees and comparing clustering patterns of trees constructed from non-synonymous mutations at codons likely to be under selection with trees constructed from synonymous substitutions at putatively neutral sites (Graser et al., 1996; Kupfermann et al., 1999; Kriener et al., 2000).

While many studies have provided evidence for TSP in MHC IIB allelic lineages, only a few have provided evidence for convergence. One study of MHC DRB sequences in New World monkeys found that PBR amino acid motifs were shared among species but some sequences of the flanking introns sorted intra-specifically (Kriener et al., 2000). Another study that favored convergence over TSP in MHC class IIB based their conclusions on different codon usage in shared amino acids among three Musteloid species, two that were sympatric and one that was allopatric (Srithayakumar et al., 2012).

While evidence for TSP in MHC genes has been found in birds (Vincek, O’Huigin & Klein, 1997; Sato et al., 2001; Richardson & Westerdahl, 2003; Alcaide, Edwards & Negro, 2007; Burri et al., 2008; Alcaide, Lui & Edwards, 2013), few studies to date have provided convincing evidence of TSP in MHC IIB genes in passerines. The strongest evidence for TSP in passerine MHC IIB genes was shown in a study of recently diverged Darwin’s finches, where several well supported interspecific allelic lineages were described (Sato et al., 2011). Well-supported interspecific clustering of variants was also reported in closely related species within the Muscicapidae family (Zagalska-Neubauer et al., 2010).

The unusually complex nature of the passerine MHC may partially explain why TSP is difficult to detect in this order. The MHC of passerines is among the most complex in terrestrial vertebrates, exhibiting highly duplicated classical MHC class I and II genes and more than ten functional class II B loci have been reported in some species. This contrasts sharply with other bird taxa such as those within the Galliformes, where one to three MHC IIB loci have been reported (Balakrishnan et al., 2010; Bollmer et al., 2010; Eimes et al., 2012; Sepil et al., 2012). It is likely that the unusually high levels of polymorphism observed in the MHC of songbirds is the result of a combination of several factors, including gene duplication, subsequent recombination (including gene conversion), and balancing selection (Hess & Edwards, 2002). Because of the potential for recombination between non-homologous loci, the role of recombination, especially gene conversion, may be exaggerated in passerines relative to other vertebrate taxa in shaping observed patterns of interspecific MHC variation between, as well as within species. The best evidence for this can be seen in phylogenetic trees of songbirds, where multi-locus MHC IIB sequences tend to cluster by species, rather than by locus (e.g., Sutton et al., 2013). This lack of trans-specific clustering of orthologous loci in passerines is thought to be due to concerted evolution as a result of gene conversion (Hess & Edwards, 2002; Edwards, Grahn & Potts, 1995).

In this study, we characterized MHC IIB loci in three closely related species of crows: jungle crow (Corvus macrorhynchos japonensis), carrion crow (C. corone orientalis), and American crow (C. brachyrhynchos). We also tested for TSP and convergence by taking advantage of the unusual arrangement of phylogeny and geographical ranges of the three species. The contemporary range of eastern carrion crows and jungle crows overlaps in East Asia (southeastern China, the Korean peninsula and the Japanese archipelago) while American crows are limited to North America (Madge & Burn, 1993; Haring et al., 2012). Mitochondrial DNA (mtDNA) phylogenies, however, place carrion crows and American crows in a monophyletic group, with jungle crows paraphyletic to these two species (Haring et al., 2012). The genus Corvus likely has a Palearctic origin, and mtDNA phylogenies suggest that carrion crows and American crows diverged in the Nearctic relatively recently; jungle crows represent a single, older lineage originating in the south-east Asian tropics (Haring et al., 2012).

Crows are suitable species for assessing TSP and convergence because their habitat, diet and social behavior are likely to render them susceptible to a diverse array of pathogens. Crows are omnivores that forage, in part, on carrion and human refuse (McGowan, 2001); scavengers, in general, could be under selection for more robust immune responses because of contact with pathogen-rich carcasses (Blount, 2003). Furthermore, all three of the Corvus species investigated in this study are found in densely human-populated areas (Ali & Ripley, 1972; Richner, 1989; McGowan, 2001; Kurosawa et al., 2003), and some data suggest that species with the ability to exploit urban areas have disproportionately strong immune responses (Moller, 2009). Finally, most crows are highly gregarious, often foraging and roosting in large communal flocks in the non-breeding season (Madge & Burn, 1993; Verbeek & Caffrey, 2002), which could promote pathogen transmission among conspecifics (Bull, 1994; Frank, 1996; Møller et al., 2001).

For this study, we assumed that the two sympatric species (carrion crows and jungle crows), which have similar habitats and co-occur in east Asia, likely share a more similar suite of pathogens with each other than with American crows. Diverse pathogen burdens are well-documented in all three species, and a comparative study of jungle crows and carrion crows revealed that ten specific helminth species were shared between the two species (Mizuno, 1984; Miller et al., 2010; Wheeler et al., 2014). There have been several surveys of intestinal parasites in American crows; however no report to date has identified these Asian helminth species in American crows, except for one anecdotal case (Jones, 1968; Hendricks, Harkema & Miller, 1969; Cawthorn, Anderson & Barker, 1980; Naderman & Pence, 1980; Mizuno, 1984; Miller et al., 2010). A strong relationship between MHC class IIB and resistance to helminth parasites has been established (Goüy de Bellocq, Charbonnel & Morand, 2008; Zhang & He, 2013). For example, Froeschke & Sommer (2012) reported both a positive and negative relationship between helminth parasites and specific MHC alleles in rodents. Thus, similar selective pressures may have acted on the MHC IIB of carrion crows and jungle crows, and a signal of convergent evolution may be detectable.

We tested for TSP among all three species of crows and convergence among carrion crows and jungle crows by analyzing a 246 bp fragment of MHC IIB exon 2 generated by 454 pyrosequencing. For the analyses, we used five different nucleotide partitions as well as supertyping, which clusters functionally equivalent amino acid sequences into units of selection (Sette & Sidney, 1999; Lund et al., 2004; Sidney et al., 2008). We also estimated species divergence times for all three species using published, as well as newly generated, mtDNA Control Region (CR) sequences.

Materials and Methods

Sample collection

We analyzed the data from 18 individuals of each species. American crows were sampled from nestlings (one bird per nest) in Yolo County, California (38°32′N, 121°45′W) on the campus of the University of California, Davis, in May and June 2012. Collection methods were approved by the Institutional Animal Care and Use Committee of the University of California, Davis (Permit Number: 16897). Approximately 50 µL of whole blood was preserved in Queens’s lysis buffer, and gDNA was extracted using a standard phenol/chloroform protocol followed by ethanol precipitation. Previously extracted gDNA from the blood of jungle crows and carrion crows was obtained from the Japanese National Museum of Nature and Science (Tsukuba, Japan) and the Yamashina Institute of Ornithology (Chiba, Japan). These samples were collected between 1994 and 2010 from the entire range of the Japanese archipelago, although most jungle crows were from the Tokyo area (16 from Tokyo, 1 each from Hokkaido and Okinawa), whereas carrion crow collection was more evenly distributed across the Japanese archipelago (6 from Kagoshima, 1 from Tsushima, 4 from Nara, 1 from Osaka and 6 from Tokyo; Fig. S1). One mL of whole blood for RNA extraction was collected from three juvenile jungle crows trapped at Ueno Zoo in Tokyo, Japan in 2012. The blood was flash frozen on site and stored at −80 °C. Jungle crow RNA was extracted from whole blood using the RNeasy Protect Animal Blood System (Qiagen, Hilden, Germany).

Amplification of MHC class IIB exon 2

In order to isolate MHC IIB exon 2, which codes for the PBR, jungle crow cDNA from two individuals (U2 and U3) was synthesized with the LongRange 2Step RT-PCR System (Qiagen, Hilden, Germany) using the manufacturer’s recommended amplification protocol. MHC IIB was then targeted in the cDNA using the primers MHC05 (Miller & Lambert, 2004), which anneals within exon 1 in several passerines (positions variable), and ComaIIbex3R, which was designed by aligning the conserved regions of IIB exon 3 from several passerines (positions 19–38). These primers amplified a 359 bp cDNA fragment. We verified that the amplicon was MHC IIB by cloning the fragment from both individuals (TOPO XL One Shot; Invitrogen, Carlsbad, California, USA). Twenty-four clones from each individual were selected and whole colony PCR was performed using the TOPO vector primers M13 F and M13 R. The 20 µL PCR contained a pluck of colony cells as template, 0.625 µM of each primer, 0.5 mM of each dNTP, 1.5 mM MgCl2, 1X PCR buffer and 0.5 U of ExTaq (Takara, Mountain View, California, USA) polymerase. The cycling conditions included an initial denaturation at 94 °C for 2 min followed by 30 cycles of 94, 55 and 72 °C each for 1 min and a final extension of 72 °C for 10 min. We sequenced each amplicon in both directions using M13 primers (Applied Biosystems 3130xl; Applied Biosystems, Carlsbad, Calfornia, USA). We performed a BLAST search and identified eight sequences that aligned with known passerine MHC IIB.

We targeted intron 1 by aligning the cDNA-derived amplicons using MUSCLE in Geneious 6.1.4 (Biomatters, San Francisco, California, USA) and a reverse primer, ComaIIbex2R, was designed in a conserved region of exon 2 and paired with MHC05. Using gDNA from two jungle crows (Coma5061 and Coma7509), a 516 bp fragment extending from exon 1 to exon 2 was amplified using 50–100 ng of gDNA, 1.0 µM of each primer, 0.5 mM of each dNTP, 1.5 mM MgCl2, 1X PCR buffer and 0.5 U of ExTaq (Takara, Mountain View, California, USA) polymerase (25 µL total volume). The cycling conditions included an initial denaturation at 94 °C for 30 s followed by 30 cycles of 94 (20 s), 60 (20 s) and 72 (45 s) °C and a final extension of 72 °C for 10 min. Each of these two amplicons was cloned and 10 colonies were sequenced in each direction using the cloning and whole colony PCR protocol described above. Sequences were aligned using MUSCLE and a forward primer flanking exon 2, ComaiF2, was designed for the (100%) conserved region at position 282–304 of intron 1. Next, we isolated intron 2 by pairing ComaiF2 with ComaIIBex3R using the same individuals and PCR, cloning, and whole colony PCR conditions described above. When amplifying from exon 2 to exon 3 in jungle crows, we discovered indels that resulted in large gaps at different locations in intron 2. 454 pyrosequencing is most efficient when sequences are of equal length, and for this reason, we designed a single degenerate reverse primer, ComaEx2RA, between positions 249–268 of exon 2. Finally, we confirmed that the primers ComaiF2 and ComaEx2RA amplified MHC IIB loci in all three crow species. Primer sequences and annealing temperatures are listed in Table S1.

454 pyrosequencing

Fusion primers were synthesized (Fasmac, Atsugi City, Kanagawa, Japan) by ligating the standard Roche multiplex identifiers (MIDs) to both the forward and reverse MHC IIB primers and the 454 adaptor primers (the Roche “Basic Amplicon” design). In order to minimize the formation of PCR artifacts, we reduced the cycle number and primer concentration and eliminated the final extension step during amplicon generation (Lenz & Becker, 2008). PCRs contained 25–50 ngs of template, 0.5 µM of each fusion primer, 0.5 mM of each dNTP, 1.5 mM MgCl2, 1X PCR buffer, 5% DMSO and 0.5 U of ExTaq (Takara, Mountain View, California, USA) polymerase (25 µL total volume). The cycling conditions included an initial denaturation at 94 °C for 30 s followed by 28 cycles of 94 (20 s), 60 (20 s) and 72 (45 s) °C.

PCR amplicons were purified using the Agencourt AMPure XP System (Beckman Coulter, Brea, California, USA) and recovered DNA was checked for quality and primer dimers on a 1.5% agarose gel. Samples were then quantified using the Quant-iT PicoGreen dsDNA Assay Kit (Invitrogen, Carlsbad, California, USA) and pooled in equimolar amounts according to the Roche GS Junior Titanium Amplicon Library Preparation Method Manual. The quality and concentration of the pooled samples was checked on a 2100 Bioanalyzer (Agilent Technologies, Santa Clara, California, USA) and then sequenced using the Roche GS Junior Titanium Sequencing System at the Japanese National Museum of Nature and Science in Tsukuba, Japan.

Amplicons were bi-directionally sequenced, and reads that passed the initial 454 Roche Junior quality filter were de-multiplexed using jMHC (Stuglik, Radwan & Babik, 2011). jMHC is especially efficient for de-multiplexing bi-directionally sequenced double-tagged amplicons because it bins only those sequences that contain both forward and reverse primers and their associated MIDs with no ambiguous characters (“N”s) in the primers, MIDs or target sequence. Sorted FASTA files (excluding primers and MIDs) were then imported into Geneious and aligned with published MHC class IIB sequences. Sequences containing indels were removed from the data set at this time.

Bioinformatics variant validation and genotyping

Variants were accepted as “true” if they occurred in at least three reads (r > 2) in both of two independent PCRs of the same individual. The criteria of (r > 2) is derived from the estimate of a read containing at least one sequence error for a 171 bp fragment of (HLA) DRB exon 2 of ∼0.11 (Galan et al., 2010). Thus, the probability that an identical error occurs three times in the same sample is ∼10−8 (Galan et al., 2010). At this stage, the most likely types of artefacts remaining in the data set are either single base substitutions or fragment chimeras generated during PCR. Although both of these artifacts are randomly generated, if these errors occur early in the PCR, they could be replicated and represented at relatively high frequencies in individual amplicons. To further reduce the probability of erroneously including single base substitutions, we classified sequences as true variants only when they differed by at least two nucleotide substitutions from more common variants. Galan et al. (2010) reported an average chimera frequency of 0.06 and some chimeras occurred at frequencies >0.1; thus, elimination of chimeras using frequency thresholds may erroneously inflate variant estimates. It is highly unlikely that two independent PCRs would each generate identical chimeras (where as many as 10 different variants could recombine) or single base substitution errors three times (r > 2). By validating variants with a second PCR for each individual (rather than a second PCR within or across individuals) chimeras are eliminated, yet variants that are rare in the population are not erroneously eliminated from the data set. The total number of reads that were included at this stage was then designated as “net reads” for the purposes of genotype confidence level calculations.

We used the program ‘Negative Multinomial’ to calculate the number of sequences that are necessary for amplifying all the variants at least three times with a confidence level of 99% (Galan et al., 2010). While the program is limited to eight possible variants, the relationship between n, net reads, r, the number of times each variant must be sampled, and m, the maximum number of variants per amplicon, is linear. Our cloned sequences indicated a maximum of 20 variants per individual which corresponds to a minimum of 175 reads for 99.9% confidence; however, this assumes equal sampling of variants. Because amplification bias is likely when co-amplifying 10 loci, we expected this bias to be reflected in over- and under-representation of some variants (Burri et al., 2014). Therefore, we increased the minimum required net reads for genotyping to 275, which represents a frequency of 1.1% (amplification bias of ∼ one order of magnitude) for a variant that has exactly three copies in an amplicon represented by 275 net reads.

Tests for selection and MHC IIB phylogenetics

For each species, we generated an alignment of validated variants, and identified the codons that have been shown to comprise the PBR of HLA in humans (Brown et al., 1993). While most MHC studies rely on the HLA PBR codons identified in humans to estimate selection, it has not been shown that these same codon positions apply universally or that selection is limited to peptide binding codons. Therefore, in order to identify specific codons that are likely under selection in crows, we performed a Wu–Kabat analysis on the amino acid alignment of verified alleles in all three species. A Wu–Kabat plot predicts amino acids that are likely subject to selection by identifying positions of high variability. Variability is calculated by dividing the number of amino acids at a position by the frequency of the most common amino acid (Wu & Kabat, 1970). For each species, we identified codons/amino acids as likely under selection if the Wu–Kabat variability metric was ⩾ twice the mean of Wu–Kabat values for all sites (Wu & Kabat, 1970; Bos & Waldman, 2006). These sites are denoted WuK-PS (PS = polymorphic site). We tested for selection on the HLA PBR sites by calculating the ratios of non-synonymous (dN) and synonymous mutations (dS) using a Z test (modified Nei-Gojobori-Jukes Cantor correction) in MEGA v. 6.0 (Tamura et al., 2011). For each species, individual codons were tested for selection using the HYPHY package in MEGA. This method, known as the counting method, estimates the number of nonsynonymous and synonymous mutations that have occurred at each codon by using a maximum likelihood reconstruction of the ancestral state of each sequence. The test statistic dN–dS is used for detecting codons that have undergone selection and for positive values the probability of rejecting the null hypothesis of neutral evolution (p-value) is calculated (Suzuki & Gojobori, 1999; Pond & Frost, 2005). Significant p values are denoted as positively selected sites. This method is more computationally compatible with large data sets than empirical or hierarchal Bayesian approaches (Yang et al., 2000), yet has been shown to provide nearly identical results with a sufficient number of sequences (Pond & Frost, 2005).

We constructed nucleotide phylogenies from five different partitions of exon 2: variable sites identified by the Wu-Kabot plot (WuK-PS, 33bp), non-WuK-PS sites (213 bp), HLA PBR sites (66 bp), non-HLA PBR sites (180 bp) and the contiguous 246 bp fragment of exon 2. For each analysis, 24 different nucleotide substitution models were tested for best fit (ML) in MEGA and the Jukes-Cantor (JC) Model + G (Gamma) + I (invariant sites) was chosen based on the Bayesian Information Criterion (BIC) score.

Supertyping

Functionally equivalent MHC IIB alleles were clustered using the supertyping method described in Sepil et al. (2012), a study that also examined highly duplicated MHC loci in a passerine species. Briefly, amino acid sites identified as positively selected sites were aligned and characterized by five physicochemical descriptor variables: z1 (hydrophobicity), z2 (steric bulk), z3 (polarity), z4 and z5 (electronic effects). These descriptor variables were placed into a matrix and subjected to K-means clustering algorithm (Sandberg et al., 1998; Doytchinova & Flower, 2005). Discriminant analysis of principle components (DAPC) was used for describing clusters in the ‘adegenet’ package in R (Jombart, Devillard & Balloux, 2010). We performed three different supertype analyses, using three different partitions: (i) the nine positively selected sites identified in the HYPHY analysis that were shared among all three species; (ii) WuK-PS sites that were shared among all three species; and (iii) HLA PBR codons.

DNA phylogenetics and species divergence

To ensure that the mtDNA phylogenies used in Haring et al. (2012) were consistent with our populations, we tested a subset of carrion crows (N = 4), jungle crows (N = 6) and American crows (N = 4) at the same mtDNA region using the primers CR-Cor+ and Phe-Cor- (Table S1) and the same PCR conditions as Haring et al. (2012), followed by Sanger sequencing as described above. We aligned these sequences with four mtDNA sequences from each species that were used in the original phylogenetic construction by Haring et al. (2012). The Hasegawa-Kishino-Yano (HKY) model (Lenz et al., 2013) had the lowest BIC scores in the best-fit substitution model (ML). HKY ML trees with 500 bootstrap replicates were constructed in MEGA v. 6.0.

Using the mtDNA sequences, we estimated divergence times for the three crow species: the divergence of carrion crow/American crow clade from jungle crow, and the divergence between carrion crows and American crows. Divergence times to the most recent common ancestor were estimated in BEAST v. 2.0 (Bouckaert et al., 2014). We used the HKY site model and a Yule process speciation prior for the branching rates. We tested the assumption of a molecular clock using the maximum likelihood method which compares the ML value for the given topology with and without the molecular clock constraints under the HKY model (MEGA v. 6.0). We ran two separate analyses using two different mtDNA substitution rates (per site, per My) previously estimated for crows (C. macrorynchos = 0.0567555 and C. coronoides = 0.0603431) (Nabholz, Glemin & Galtier, 2009). The Markov chain Monte Carlo (MCMC) analyses were run for 107 generations (10,000 iteration burn in); the mean and 95% highest posterior density interval (HPD) for divergence times were calculated in Tracer v. 1.6 (Bouckaert et al., 2014).

Results

Pyrosequencing

The goal of the first 454 run was to estimate the number of IIB loci and the minimum read number required for accurate genotyping (99.9% confidence). Six individuals from each species were sequenced and a total of 125,839 reads passed the initial 454 filter. After excluding reads of less than 270 bp, there were 76,891 reads with an average length of 340 nucleotides. Amplicons had between 8,384 and 483 initial reads (mean = 1,213) that were sorted by jMHC. These samples had an average of 376 net reads (range 171–3,211). The maximum number of variants (differing by at least 2 nucleotides) found in a single individual was 20, which suggests that crows have up to 10 IIB loci per haplotype. The required minimum read number was calculated as 211; however, we increased the cutoff to 275 net reads to account for amplification bias (see methods).

For the second 454 run, a total of 107,279 reads passed the initial 454 filter. After excluding reads of less than 270 bp, there were 90,203 reads with an average length of 340 nucleotides. A total of 18 individuals (with replicates) from each species met the criteria for variant validation of at least 275 net reads and at least three identical reads per variant in both of two independent PCRs of the same individual. All DNA sequences were deposited in NCBI Genbank: Accession numbers: MHC IIB: TBA; crow mtDNA: KM246294–KM246308; crow nuclear intronic: KM246309–KM246323

MHCIIB variation and selection

The 248 bp fragment was conserved at the final two base pair positions in all three species and was trimmed to 246 bp. The number of validated MHC IIB nucleotide variants for each species was: jungle crows = 89, carrion crows = 81 and American crows = 67. Individuals had a range of 7–20 variants, indicating that the MHC IIB in crows exhibits copy number variation (Table 1), although it is possible that the range of variant number could be explained by other phenomena, including null alleles, allele sharing across loci, homozygous vs. heterozygous loci or the variant validation methodology (where sequences with indels were excluded from the data set). Few variants were shared among species and there was no discernable pattern of allele sharing relating to phylogeny or sympatry: Four unique variants were shared among American crows and jungle crows, three among American crows and carrion crows, three among carrion crows and jungle crows and four variants were shared by all three species.

Table 1 MHC IIB variation and historical selection in three species of crows.

Species	Alleles (total)	Alleles (Indiv)	π	π SPBR	π NPBR	dN/dS PBR	p value Z-test	
Jungle	89	7–18	0.16	0.29 ± 0.115	0.47 ± 0.09	1.6	0.09	
Carrion	81	11–20	0.15	0.24 ± 0.02	0.45 ± 0.1	1.9	0.028	
American	67	11–18	0.14	0.25 ± 0.02	0.44 ± 0.09	1.6	0.047	
Notes.

N 18 for each species

π nucleotide diversity across 246 bp of exon 2

π SPBR synonymous mutation rate

π NPBR non-synonymous mutation rate at the HLA PBR identified by Brown et al. (1993)

Z-test significance (p) value using only HLA PBR codons.

The codon-based Z test for selection (non-synonymous/synonymous mutations) averaged across the PBR yielded a significant p value for American crows and carrion crows but not for jungle crows (Table 1). The Wu-Kabot plot identified 11 amino acid sites that were highly variable in all three species: 6, 23, 25, 32, 33, 52, 61, 62, 65, 73 and 81. Eight of these sites (73%) corresponded to known peptide binding sites in HLA (Fig. 1). Among the three species, jungle crows exhibited the most amino acid variation, with five hypervariable (>20 Wu–Kabat index) amino acid sites (Fig. 1). The HYPHY test for selection identified nine positively selected sites that were shared among the three species (2, 4, 33, 52, 55, 66, 68, 72 and 81), but only 5 of these (56%) corresponded to the HLA PBR (Fig. 1).

Figure 1 Wu–Kabat plot of amino acid sequences from a 246 bp fragment of MHC IIB Exon 2 for three species of crows.

Blue circles represent HLA PBR codons identified by Brown et al. (1993). Red ovals identify positively selected sites (PSS) identified by HYPHY. Variability is represented on the Y axis and site position is on the X axis. Histogram key: Red, jungle crow; yellow, carrion crow; blue, American crow.

In total, we recovered 172 unique amino acid sequences among the three species (70 in jungle crows, 58 in carrion crows and 44 in American crows). We detected no species-specific motifs or motifs that were shared exclusively between two species. Within-species amino acid diversity was relatively high (0.245 ± 1.5), similar to that which we calculated for scrub jays (Aphelocoma coerulescens), another corvid species (23.3%; N = 11; NCBI (U23958-65; 72,73,75) (Edwards, Grahn & Potts, 1995). An alignment (82 sites) of common amino acid fragments from the three crow species with four other passerine species, and a falconiform species for comparison, highlights the high level of amino acid diversity found among and between species of passerines (Fig. S2).

MHC IIB phylogenies

We observed a similar pattern among the five different partitions of exon 2: MHC IIB variants formed several supported (bootstrap values ≥50), interspecific clusters in trees comparing putatively functional sites. Interspecific clusters were also observed when comparing putatively neutral sites, but there were fewer of these supported clusters, and in general they were less inclusive. This pattern was most striking in trees constructed using the contiguous 246 bp exon 2 fragment (Fig. 3). For instance, a tree that was constructed by comparing non-synonymous substitutions (Fig. 3A) shows six well supported clusters, each containing all three crow species, whereas a tree constructed by comparing synonymous substitutions of the same fragment (Fig. 3B), shows only three such clusters. Supported clustering at putatively functional sites was also more inclusive in these exon 2 trees, with 44% of all variants (105/237) forming supported interspecific clades compared to just 18% (32/237) in the tree using synonymous sites. The trees constructed from WuK-PS and non-Wuk-PS sites (Figs. S3A and S3B) as well as those constructed from the HLA PBR and non-PBR sites (Figs. S4A and S4B) displayed similar, though less distinct patterns. Among all partitions, the most common (supported) clustering was among all three species. We did not observe biased clustering of variants among the sympatric carrion crows and jungle crows; or among the more recently diverged, allopatric, American crows and carrion crows.

Figure 2 Maximum likelihood tree analyzing the mtDNA control region of jungle crows, carrion crows and American crows.

Numbers below the branches are the bootstrap support values for each clade, and numbers above the branches are the estimated species divergence time in millions of years (Ma), followed by the 95% highest posterior density interval (HPD) from MCMC simulations performed in BEAST v. 2.0. The mtDNA mutation rate used in the divergence time estimate was 0.0567555 (Nabholz, Glemin & Galtier, 2009). Accession numbers with an asterisk are from this study; all others are from Haring et al. (2012). Coco, carrion crow; Cobr, American crow; Coma, jungle crow. Aphelocoma coerulescens (Accession number AF218919) was used as an outgroup.

Figure 3 Neighbor joining trees of MHC IIB Exon 2 variants in three species of crows.

The nucleotide fragment was 246 bp. Branches with bootstrap support (Jukes Cantor, 500 replicates) are indicated: light red ≥50%; dark red ≥70. Coco (orange), carrion crow; Cobr (blue), American crow; Coma (green), jungle crow. Supported clades representing all three species are denoted by asterisks. Scale bar indicates substitutions per site. (A) Tree constructed by comparing non-synonymous substitutions/nonsynonymous site. Red shading indicates supported interspecific clades. (B) Tree constructed by comparing synonymous substitutions/synonymous site. Blue shading indicates supported interspecific clades.

Supertyping

Among the three species, there were nine shared positively selected sites in exon 2. Five of these nine sites corresponded to HLA PBR sites (Fig. 1). From the 140 unique positively selected sites amino acid variants a total of eight supertypes were identified (Fig. 4). Only one supertype (Supertype 4) was limited to carrion crows and jungle crows, whereas the rest of the supertypes were shared among the three species. The supertypes generated by the WuK-PS and the HLA PBR codons were not well supported and thus excluded from our analysis.

Figure 4 Discriminant Analysis of Principle Components (DAPC) scatterplot of the 8 MHC supertypes.

10 principle components (PC) and three discriminant functions (dimensions) were retained during analyses, to describe the relationship between the clusters. The scatterplot show only the first two discriminant functions (d = 2). The bottom graph displays the bar plot of eigenvalues for the discriminant analysis. The Eigenvalues that were not retained are marked in white. The Eigenvalues that are used in the scatterplot are marked in black. The Eigenvalue that was retained for the analysis but not used in the scatterplot is marked in grey. Each allele is represented as a dot and the supertypes as ellipses.

mtDNA phylogeny and divergence estimate

The CR-Cor+ and Phe-Cor- primers amplified an 836–920 bp fragment of the mtDNA CR region in the three species of crows. The ML tree placed carrion crows and American crows in a separate clade from jungle crows with 100% bootstrap support (Fig. 2), thus confirming that the jungle crows and carrion crow populations from this study conform to the phylogeny of Haring et al. (2012). The ML-based test of the molecular clock failed to reject the null hypothesis of equal evolutionary rate throughout the tree (p = 0.85); thus we used a strict clock setting for the MCMC iterations in the BEAST program. Using the mtDNA substitution rate of 0.0567555/site/My the estimated mean divergence time to MRCA of all species was 0.8992 Ma with a 95% HPD interval of 0.7055, 1.1147 Ma (effective sample size (ESS) = 8581.7, median = 0.8942, standard deviation = 0.107). The estimated divergence to the MRCA of carrion crows and American crows was 0.4372 Ma with a 95% HPD interval of 0.3036, 0.5801 Ma (ESS = 7682.7, median = 0.4319, standard deviation = 0.0723) (Fig. 2). Divergence estimates using the higher substitution rate (C. coronoides = 0.0603431/site/My) yielded similar values: the MRCA of all species = 0.8509 Ma, HPD interval = 0.6578, 1.0486 Ma; MRCA of carrion crows and American crows = 0.4143 Ma, HPD interval = 0.2881, 0.5458 Ma.

Discussion

We observed strong interspecific clustering of MHC IIB variants in three closely related species of crows (Corvus sp.), and most of the supported clusters contained all three species; however, the clustering was much more inclusive, and better supported, when comparing non-synonymous substitutions at putatively selected sites than synonymous substitutions at putatively non-selected sites (Fig. 3; Figs. S3 and S4). This pattern was observed in all of the phylogenies constructed from the different partitions of exon 2, but was most obvious in the trees constructed using the entire 246 fragment of exon 2 (Fig. 3). This pattern of strong clustering among functional sites is suggestive of convergent evolution (Klein, Sato & Nikolaidis , 2007). In fact, if this study been limited to the sympatric carrion crows and jungle crows, we may have erroneously concluded that convergent evolution explains these patterns; however, inclusion of sequences from the allopatric American crow, and the resulting interspecific clustering of all three species at functional sites, suggests that the pattern observed in the phylogenies is not convergence, but rather trans-species polymorphism.

Our results indicate that the MHC IIB in crows exhibits the same characteristics of the MHC IIB observed in other passerine species (Bollmer et al., 2010; Zagalska-Neubauer et al., 2010), namely, high levels of polymorphism, highly duplicated loci, probable copy number variation of loci and a strong signature of selection. Consistent with similar studies, we also observed highly polymorphic amino acid positions and detected selection at sites other than the HLA PBR (Fig. 1), which indicates that either the PBR in passerines comprises different codon positions than the human MHC or that regions other than the PBR are under selection (Anmarkrud et al., 2010; Bollmer et al., 2010; Zagalska-Neubauer et al., 2010). The Wuk-Kabot plot was a better fit to the HLA PBR than the HYPHY probability based test for selection (Fig. 1), which may indicate that such probability tests are not reliable when identifying selected sites across many duplicated loci if the relationships between these loci are unknown (e.g., orthologous versus paralogous relationships). For example, it is unlikely that the hyper-polymorphic amino acid sites at positions 6, 23, 32, 62, and 65 are not subject to selection. This highlights the importance of using a combination of different exon 2 partitions when evaluating selection or establishing putative neutral and selected sites for phylogenetic analyses in complicated systems such as the passerine MHC.

Several factors may explain the relatively high MHC IIB neutral genetic variation between allelic lineages we observed in these crow species. First, if these allelic clusters represent ancient lineages, they are likely millions of years older than the MRCA of all three species, which allows sufficient time for random mutations to be incorporated, and the likelihood that these neutral mutations persist is increased by their close association with selected sites (Klein et al., 1998; O’hUigín et al., 2000). Thus, the majority of the synonymous polymorphisms observed in neutral sites were likely inherited, rather than generated since species divergence. Other factors that may contribute to the relatively high synonymous nucleotide variation between species observed at non-selected sites relative to selected sites are the combined effects of recombination through genetic exchange and gene conversion. In a study of MHC IIB in sticklebacks (Gasterosteidae), Lenz et al. (2013) observed a similar pattern of clustering of selected sites (in this case, the HLA PBR) but a lack of clustering in the remaining sites of exon 2, a pattern similar to the one reported here. The authors reasoned that selected sites may evolve under site-specific functional constraints that maintain similar amino acid motifs, and that any new variants generated by recombination are more likely to be retained if they share this functional similarity. Intraspecific recombination and gene conversion could then erode the signal of common allelic ancestry over time.

Gene conversion is thought to explain why passerine MHC sequences often sort by species, rather than by locus, on phylogenetic trees (Hess & Edwards, 2002). Elevated rates of gene conversion have been documented in birds, and one study reported a gene conversion rate over a magnitude higher than the neutral mutation rate (Spurgin et al., 2011). It is possible that the reason why the crow MHC IIB sequences do not sort intra-specifically is that insufficient time has passed since species divergence for gene conversion to homogenize the neutral sites within exon 2. The patterns we observed were similar to those described for MHC IIB in Darwin’s finches in that the sequences sorted in a manner that would be expected from a single, undifferentiated species (Sato et al., 2011). The similar results between our study and that of Sato et al. (2011) are likely due to the relatively recent divergence times of the focal taxa in both studies. In the Galapagos, Darwin finches’ radiated from a founding population approximately 2.5 Ma, while the MRCA to the three crow species was estimated at approximately 900,000 years ago.

While we did not detect evidence of convergence between the sympatric carrion crow and jungle crow in the MHC IIB phylogenies, the supertype analysis revealed a pattern consistent with convergent evolution. While seven of the eight supertypes were shared among the three species, one supertype, ST-4, was exclusive to carrion crows and jungle crows, which might reflect selective pressure from a specific, shared parasite. None of the other seven supertypes were exclusive to either a single species or group of two species. This supertyping method classifies MHC II B loci into clusters based on the physio-chemical properties of the amino acids in the positively selected sites of the PBR. Several studies of primate MHC have shown that different MHC alleles bind very similar peptide motifs, suggesting significant overlap in peptide binding repertoires (Sette & Sidney, 1999; Lund et al., 2004; Sidney et al., 2008). Thus, different amino acid sequences can be functionally equivalent, and grouped together as a supertype, considered a unit of selection. Recently, Sepil et al. (2013) showed that different MHC class I supertypes confer resistance to malaria in great tits (Parus major). In our study, supertypes ST-3 and ST-4 grouped together, away from the other six supertypes, which may suggest that these groups are functionally important for a specific type of parasite, whereas the other supertypes may confer resistance to more common, generalist parasites (Fig. 4).

Conclusion

In the present study we describe strong evidence for trans-specific polymorphism at the MHC IIB in three species of crows. Our results are consistent with two other studies that examined MHC IIB in closely related passerine taxa and indicate that TSP in passerines may be common, but due to intra-specific gene conversion, most easily detectable among recently diverged species.

Given that the allopatric American crow and carrion crow diverged approximately 430,000 years ago, we expected to find some signal of convergence of MHC IIB genes between the sympatric jungle crow and carrion crow in the MHC IIB phylogenies. While we did find patterns consistent with convergence in the phylogenetic analyses, any convergence would have had to occur among all three species, before the MRCA split approximately 900,000 years ago. While this scenario cannot be ruled out, the results presented here are better explained by TSP.

Supplemental Information

Figure S1 Map of Collection Sites in Japan

Samples numbers from each location are indicated: carrion crows, orange; jungle crows, green.

Click here for additional data file.

Figure S2 MHC IIB Exon 2 Amino Acid Alignment

Common alleles (occurring in ≥4 birds) of three species of crows (Coma: Corvus macrorhynchos, Coco: C. corone, Cobr: C. brachyrhynchos) with three other species of Passeriformes (Apco: Aphelocoma coerulescens, Getr: Geothlypis trichas, Tumi: Turdus migratorius) and one representative of Falconiformes (Fati: Falco tinnunculus). Accession numbers for crows are [TBA]. Accession numbers for other species are listed after label.

Click here for additional data file.

Figure S3 Neighbor Joining Trees of Wu-Kabot Plot MHC IIB Variants

Branches with bootstrap support (Jukes-Cantor, 500 replicates) are indicated: light red ≥50%; dark red ≥70. Coco (orange), carrion crow; Cobr (blue), American crow; Coma (green), jungle crow. (S3A) Tree constructed by comparing non-synonymous substitutions/nonsynonymous site from codon positions identified as variable from a Wu-Kabot Plot (11 codons). Red shading indicates supported interspecific clades. (S3B) Tree constructed by comparing synonymous substitutions/synonymous site at remaining (non-Wu-Kabot) sites (71 codons). Blue shading indicates supported interspecific clades.

Click here for additional data file.

Figure S4 Neighbor Joining Trees of MHC IIB PBR Variants

Branches with bootstrap support (Jukes-Cantor, 500 replicates) are indicated: light red ≥50%; dark red ≥70. Coco (orange), carrion crow; Cobr (blue), American crow; Coma (green), jungle crow. (S4A) Tree constructed by comparing non-synonymous substitutions/nonsynonymous site from codon positions identified as the peptide binding region (PBR) in HLA (Brown et al., 1993) (22 codons). Red shading indicates supported interspecific clades. (S3B) Tree constructed by comparing synonymous substitutions/synonymous site at remaining (non-PBR) sites (60 codons). Blue shading indicates supported interspecific clades.

Click here for additional data file.

Table S1 Primers used to amplify MHC IIB cDNA and gDNA in Crows

Primers ComaIIbex2r and ComaEx2RA were ligated to fusion primers for 454 sequencing.

Click here for additional data file.

We wish to Naoyuki Takahata and Colm O’hUigín for advice during the experimental phase of the project and Peter Dunn for advice on the analysis and interpretations of results. We also thank Jon Fong for assistance of the phylogenetic analyses.

Additional Information and Declarations

Competing Interests

Author Contributions

Animal Ethics

Field Study Permissions

DNA Deposition

Isao Nishiumi is an employee of the National Museum of Nature and Science, Tsukuba, Japan.

John A. Eimes conceived and designed the experiments, performed the experiments, analyzed the data, contributed reagents/materials/analysis tools, wrote the paper, prepared figures and/or tables, reviewed drafts of the paper.

Andrea K. Townsend analyzed the data, contributed reagents/materials/analysis tools, wrote the paper, reviewed drafts of the paper.

Irem Sepil performed the experiments, analyzed the data, contributed reagents/materials/analysis tools, wrote the paper, prepared figures and/or tables, reviewed drafts of the paper.

Isao Nishiumi performed the experiments, analyzed the data, contributed reagents/materials/analysis tools, reviewed drafts of the paper.

Yoko Satta analyzed the data, contributed reagents/materials/analysis tools, reviewed drafts of the paper.

The following information was supplied relating to ethical approvals (i.e., approving body and any reference numbers):

For American crow only: This work was performed under protocols approved by the Institutional Animal Care and Use Committee of the University of California, Davis (Permit Number: 16897). All work was conducted on the private property of the University of California, Davis. No protected species were sampled. Carrion and jungle crow samples were from museums collections.

The following information was supplied relating to field study approvals (i.e., approving body and any reference numbers):

For American crow only: This work was performed under protocols approved by the Institutional Animal Care and Use Committee of the University of California, Davis (Permit Number: 16897).

The following information was supplied regarding the deposition of DNA sequences:

NCBI Genbank Accession numbers:

MHC IIB: KP888319–KP888555;

Crow mtDNA: KM246294–KM246308.

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
