# Peer review of "Patterns of evolution of MHC class II genes of crows (Corvus) suggest trans-species polymorphism"

_PeerJ, doi:10.7717/peerj.853_

## Round 0.1 · original submission · Major Revisions

I have now received three very thorough reviews. Although all three reviewers liked the paper, several mentioned areas of ambiguity and reviewer 3 in particular suggests that the hypotheses may need to be clarified, re-stated, or presented with caveats. For example, you may want to indicate that the hypotheses presented may be somewhat arbitrary and do not cover the full range of possible scenarios. Additionally, it is very important to mention the potential roles of recombination and gene conversion in generating and complicating MHC evolution. Ideally you could use a program like omegaMap, which takes recombination into account while estimating positive selection. Some mention of recombination should be made.

Reviewer 1 ·

Basic reporting

The authors provide a clear synopsis of MHC gene evolution in birds, and relevant background information regarding their focal species with appropriate references cited throughout.

The aims and primary hypotheses are largely presented in a clear manner; however, the statement on lines 106-107 that ‘MHC phylogeny should roughly mirror phylogenies constructed from neutral markers’ if time since speciation is insufficient to allow ‘new patterns within the MHC sequences to have emerged’ (line 105) seems ambiguous. Do the authors mean that each MHC gene duplicate should individually mirror the neutral phylogeny (i.e. one clade per locus, with representatives from each species?). This hypothesis would seem to conflict with the earlier statement that ‘the role of recombination, especially gene conversion, may be exaggerated in passerines’ (lines 39-40) and would also disregard the potential role of recent (post-speciation) gene duplication? Alternatively, is it meant that alleles would largely cluster by species, regardless of locus affiliation, and therefore ‘reflecting differentiation at the time of species divergence’ (line 394) indicates that a pattern of concerted evolution would already have been apparent? More precise wording could help clarify this major hypothesis.

Fig. 1: lines 335-337 of the text state that there are ‘nine positively selected sites that were shared among the three species, but only 6 of these correspond to HLA PBR’. Fig. 1 seems to show only 5 positively selected sites that are also PBR (sites 4, 33, 55, 66, 81)?

Fig. 3: The presentation of this figure is not particularly effective, and some branches are missing (e.g. left-most branches in columns 3 & 4). Perhaps a circularized (rather than linear) tree would be more appropriate, with different colours used for taxon labels of each species (i.e. to better display the overall pattern seen for each species?). It might also be advisable to collapse nodes with < 50% bootstrap support, and a scale bar indicating the number of substitutions per site should be included.

Figs. S5 & S6 are extremely small, although they are provided at sufficient resolution for readers to zoom in to read taxon labels.

Supplemental tables and figures: Informative captions or headers should be provided.

Consistent nomenclature should be maintained throughout (e.g. use of Cobr/Coco/Coma in Figs. 3, S3, S5 & S6 versus Cb/Cc/Cm in Figs. S2, S4).

The authors might wish to reconsider several wording choices:
Line 44: ‘homologous loci’. Is ‘orthologous’ (as opposed to ‘paralogous’) what is really meant?
Line 63: ‘horizontal transfer’ might be confusing due to connotations of horizontal gene transfer. ‘…promote pathogen transmission among conspecifics’ might be preferable.
Line 169: ‘gene specific primers’ implies that individual MHC IIB loci were amplified, rather than simultaneous amplification of all paralogs.
Line 255: ‘ensure’ rather than ‘insure’

Define all abbreviations at instance of first use (e.g. ‘PBR’ on line 133 should be introduced on line 11).

Please ensure that all references are complete (e.g. Jombart et al., Nabholz et al., Savage et al., Sawai et al., Sepil et al. missing volume and/or page #s).

Experimental design

The authors thoroughly document their methodologies. The procedures of Galan et al. (2010) are used to validate true MHC IIB variants and to estimate the data requirement to achieve a high level of confidence in the resultant genotypes, and MHC IIB genotyping of all individuals is performed in duplicate. Protocols to minimize the formation of PCR artefacts are adopted.

However, it is unclear whether the methods of Galan et al. were applied in their entirety (i.e. were all reads that showed frameshift mutations removed from the dataset, as this action could affect estimates of gene copy number).

It is also not explicitly stated in the results whether all individuals were assigned identical genotypes from each replicate.

Could the inference of allele sharing across crow species be influenced by the moderate sample size within each species? In other words, how representative is sampling likely to be relative to the total variation within species? This issue could perhaps be addressed by considering whether the discovery of new variants begins to plateau with an increasing number of samples such that it appears likely that at least the most common alleles within each species have been identified.

American crow sampling occurred in a single locality and season. Jungle crows were also largely from a single region, while carrion crows were sampled from a much larger area, and both were sampled across a 16-year timescale. Is there any information about spatiotemporal dynamics of crow parasites, or of population structure of carrion crows, that might be relevant to the conclusions drawn from this sampling (i.e. factors such as pathogen diversity/local adaptation of hosts)?

Validity of the findings

MHC IIB gene number & copy number variation:
It is inferred that there ‘are 10 IIB loci in crows’ (lines 311-312), but is also stated that ‘the MHC IIB in crows exhibits copy number variation’ (line 324). In the first instance, it would appear more precise to state that there is evidence for up to 10 loci per haplotype.

The claim regarding copy number variation is probably reasonable, and is in line with previous findings in passerines; however, it should be stated as an inference rather than a conclusion since factors such as null alleles, homo- versus heterozygosity at each locus, sharing of alleles across loci, and discarding of variants with frameshifts during the validation process could all contribute to inference of gene copy number.

MHC IIB phylogeny:
On line 378, it is stated that ‘there was a general trend for crow MHC IIB sequences to cluster by species’, and on line 404 it is concluded that ‘sequences mostly clustered by species overall’. This pattern only applies to the shallowest nodes of the tree, and is not descriptive of the overall pattern within each species (i.e. there is no tendency for MHC variants within a species to all cluster together with high support).

Lines 248-249. It appears that phylogenies were constructed from subsets of non-synonymous sites, synonymous sites, PBR codons, and positively selected codons, as well as the entire 248 bp fragment of exon 2. However, aside from Fig. 3 which states that non-synonymous sites were used, it is not clear which supplemental figure corresponds to which dataset.

Can the authors detail their choice in presenting the non-synonymous dataset (rather than the entire exon) in the main text? The authors should also provide details regarding alignment lengths for each dataset, as it would appear that some phylogenies would be built from a very small number of sites.

Selection acting on MHC IIB loci:
Only 6 (or 5?) of the 9 sites identified as positively selected in crows coincide with PBR sites in human HLA, and it is suggested that ‘the PBR of crows is not identical to that identified for HLA or that sites other than PBR are under strong selection in crows’ (lines 337-338). Are these non-PBR sites also positively selected (or at a minimum highly variable) in other passerines?

The conclusion that ‘trans-species polymorphisms in the MHC IIB are maintained by balancing selection in recently diverged species of crows’ (lines 465-466) seems to be drawn largely from the results of tests for positive selection in addition to the large number of total variants that were detected. However, consideration should be given to the fact that analyses are based upon alleles shared across all loci and that individual loci might exhibit heterogeneous levels of variation and distinct evolutionary dynamics and patterns of selection.

A dichotomy is drawn between ‘neutral molecular events, such as gene conversion and recombination, or pathogen-mediated selection’ (lines 48-49), which seems to be the underpinning of the two competing hypotheses presented. However, should gene conversion and recombination not be viewed as mutational mechanisms (in addition to substitutional events) that give rise to new sequence variants that may also be subject to pathogen-mediated (or other) forms of selection?

Supertypes:
The finding of 2 supertypes that are more distinct from the remainder, and of one supertype that is shared by only jungle and carrion crows is of interest and probably merits deeper consideration.

Is a similar pattern found if supertypes are inferred from the entire predicted PBR, rather than positively selected sites only? Would it also be of interest to consider the proportion of alleles for each species that belong to each supertype? Can the major physiochemical differences that distinguish supertypes 3 & 4 be identified (i.e. what is the major loading of the 5 physiochemical descriptors for the first two principal components?).

Lines 420-422: Is there published evidence to support that supertypes identified as ‘distinct’ by clustering, yet less differentiated from other supertypes (e.g. supertypes 1, 2, and 5-8 versus 3 & 4) would necessarily correspond to alleles that recognize generalist parasites?

It seems that the analysis of supertypes also forms the basis for the conclusion that highly duplicated passerine MHC IIB loci might ‘be less specific than taxa that express few MHC loci’ (lines 459-460). This is an interesting idea and likely merits more thorough consideration (and support) within the main body of the text.

Estimation of divergence dates:
The authors clearly state that ‘generating a highly accurate estimate of species divergence time in crows was not the aim of this paper’ (lines 264-265). Nevertheless, these analyses occupy a fair amount of the manuscript and several problematic issues should be addressed.

A molecular clock is assumed throughout; however, several recent studies have concluded that there exists substantial rate variation among avian lineages for both nuclear (e.g. Lanfear et al. 2101 [PNAS 107:20423-20428]; Nabholz et al. 2011 [Mol Biol Evol 28:2197-2210]; Nam et al. 2010 [Genome Biol 11:R68]) and mitochondrial DNA (e.g. Nabholz et al. 2009 [BMC Evol Biol 9:54]). The authors should provide references to justify their statement that ‘unlike the mtDNA substitution rate, the nuclear gene mutation rate is not expected to be highly variable between taxa, and is therefore probably more reliable’ (lines 372-373).

Of particular relevance to the current study, passerines have been found to display elevated rates relative to other lineages (although the American crow showed a lower rate than other passerines in protein-coding exons in the study of Nabholz et al. 2011). The authors should therefore justify the assumption of a molecular clock and the use of a substitution rate for nuclear DNA from galliforms (rather than e.g. from the zebra finch or other passerine, or from minimum and maximum values reported across all avian lineages).

The authors should also justify the choice of the galliform estimate from Sawai et al. 2010 (1.17 x 10-9 substitutions per site per year) estimated from 30 nuclear introns rather than (or why not used in addition to) the estimate of 1.91 x 10-9 substitutions per site per year derived from divergence at fourfold degenerate sites across 8,384 1:1 orthologs by Nam et al. 2010.

The estimates of divergence dates from mitochondrial DNA might be less problematic as they are based upon data from three congeneric crow species, including one of the focal taxa. However, there still exists a 6.1% percentage difference between the fastest and slowest of these rates (C. macrorynchos at 0.0567555 versus C. coronoides at 0.0603431; Nabholz et al. 2009).

At a minimum, it would seem wise to test the assumption of a molecular clock (e.g. with a likelihood ratio test), and/or to show the recovered nuclear and mtDNA phylogenies with relative branch lengths to allow readers to assess how this assumption might impact the results. It might even be ‘better not to try estimating absolute divergence times at all and instead rely on a relative (uncalibrated) molecular time scale’ (Sauquet 2013 [C. R. Palevol 12:355-367]). If absolute divergence times are felt to be essential, they should at least be presented with 95% confidence intervals whenever possible (e.g. as could be done for the mtDNA estimates).

The authors state that ‘d within species was 0’ (line 364) for the five nuclear introns, but Fig. S3 shows non-zero terminal branches for both Cobr and Coco?

Additional comments

Overall, this study adopts appropriate methods for genotyping of duplicated MHC class II loci in three species of crows, and proposes some interesting ideas such as the relationship between gene copy number and specificity of pathogen recognition. However, some points could benefit from a deeper investigation as outlined above and methods for the estimation of divergence dates should be reconsidered or more thoroughly justified.

The authors should rigorously error-check the manuscript. For instance:
Line 146: ‘31310xl’ (3130xl)?
Line 409: convergent, not convergence
Line 416: repertoires, not repertories
Line 461: mediated, not mediates
Line 463: jungle and carrion crows (‘crows’ missing)
Fig. 1 caption: HYPHY, not HYFY
Fig. 2 caption: Nabholz et al. 2009, not 20009

Reviewer 2 ·

Basic reporting

In this paper, Eimes et al. examine the effect of pathogen-mediated selection on MHC sequence diversity using populations of three species of crow. Their dataset includes two closely related, but geographically separated species, as well as two more closely related but sympatric species. Overall I found this to be a well-presented manuscript, and the results are interesting.

I note a few points where the writing could be clarified:
The Introduction would benefit from a little more detail on the mechanism by which “balancing selection” may maintain cross-species polymorphisms, as is the key focus of this study. Are there other mechanisms that maintain such diversity?
L43-46: I found this sentence long and confusing, suggest rephrasing
L51: Please provide full genus here (I think this is the first mention of the species, outside of the Abstract)
L83: “We could find only a single case of any of these specific helminths in North America” – do you mean in American crows? If so, then this sentence is inconsistent with the one before, stating there were no reports of these helminths in American crow. Please clarify.
L91: “that occurred since both that divergence as well as” – this is difficult wording, suggest rephrasing this sentence.
L139: typo? “MHCB IIB” should be “MHC IIB”
L200 (and elsewhere): do you mean “erroneously”, rather than “errantly”?
L259: why was it necessary to clone the mtDNA sequences?
L322: please remind the reader here of the method used to “validate” an variant
L342: reference needed for the scrub jay work
L409: typo – “convergent”
L440-441: I think there is a word missing from the first part of this sentence.
L461: typo – “mediated”

I also have two suggestions for improving the presentation of figures:
Figure1: needs axis labels. Also, I believe the caption has swapped the description of x and y axes.
Figure3: I found this tree quite difficult to interpret, because it is so large. I understand that it is challenging to present such a long figure on a single page – I would suggest extending the branches that join different panels together, so that it is clear how the panels join onto one another. Would it also be possible to colour-code labels that apply to each of the three species, so that the reader can more-easily differentiate how alleles from the different species cluster together.
In addition, I could not find full captions for the Supplementary Material – this is particularly problematic for the Supplementary Figures. The titles provided on the files do not provide sufficient detail for the figures to “stand alone”, and I would suggest more detailed captions for this material.

Experimental design

The data and study design used here are suitable for addressing the questions posed. The methods and their reporting are generally appropriate and thorough. I have two comments in this respect:

1. In the Introduction, it is noted that “we attempted to detect recent pathogen-mediated selection by comparing sequences between more recently diverged allopatric species (carrion and American crows) and more distantly related sympatric species (carrion and jungle crows)” (L98-101), but I also note that in this paper “recent selection” is defined as “that occurring after speciation” (L93). It is difficult to see how post-speciation processes can be inferred by making comparisons between species. Perhaps this is simply a wording issue, but it warrants clarification.

2. Regarding the model used to test for positive selection (L238): only one model was used, although multiple possible sequence-based tests are available and similar studies often opt to use a conservative “consensus” approach. Regardless, the authors should describe the method they use (e.g. provide a reference for the method used by MEGA) and justify the choice. Would the results of this study change if a different test was used, such as the mixed effects model of evolution (MEME), which is designed to target both episodic and pervasive positive selection? If different sites in the sequence are under selection in the different crow lineages (i.e. “episodic selection”), would this confound the supertyping analysis (which was based on amino acid properties at positively selected sites detected at the whole-tree level)?

Validity of the findings

The conclusions of this paper are supported by the results. The Discussion is well-written.

Additional comments

Overall I enjoyed reading this manuscript, and believe that it will make a good contribution to our understanding of the selective processes operating on MHC diversity in natural populations.

Reviewer 3 ·

Basic reporting

No comments

Experimental design

No comments

Validity of the findings

In the current manuscript Eimes and colleagues isolated MHC class IIB (MHC IIB) genes in three species of crows. Based on a species sampling, comprising one recently diverged species pair (carrion/American crow) and one sympatric species pair (carrion crow/jungle crow), they aim at investigating the time scale at which selection acts to maintain MHC IIB polymorphisms. They hypothesize that 1) with ancient selection exclusively, the phylogenetic relationships among MHC IIB sequences should reflect the species tree, and 2) under strong convergent, pathogen-mediated selection, MHC IIB sequences should be more similar among the sympatric species than among the allopatric sister species. The results reflect none of the predicted phylogenetic patterns, and the authors conclude that their results indicate trans-species polymorphism. Furthermore, based on the presence of one common supertype among the sympatric species, they conclude on convergence.
I think that the authors chose a good sampling of species to demonstrate convergence. However, I am not convinced in how far the same sampling can provide information on the time scale(s) at which selection acts on MHC. For the latter, I think that details on the demographic history of the study species would be required and implemented in a complex modelling framework. Most importantly, I fear that there is a number of problems with the hypotheses forwarded by the authors, as well as with the interpretation of the results. I am therefore sorry to recommend the manuscript to be rejected for publication in PeerJ in its current form.

The hypotheses seem to some extent arbitrarily chosen, as more (and probably more accurate/probable) hypotheses can be made. One important point is that, contrary to what these hypotheses assume, selection and drift are not the only forces shaping MHC diversity. Recombination and gene conversion have a strong impact on MHC IIB exon 2. I think that moreover the first prediction, besides not being precise (see detailed comments), is not realistic. Why would selection only act before but not after the species split? Independent of this, the assumption that in this case the MHC topology should reflect the neutral nuclear topology is not correct. Alone the action of selection would result in a completely different coalescent at the MHC compared to neutral markers, as it leads to different ancestral population sizes of the two types of markers. Under balancing selection, more incomplete lineage sorting would be expected at the MHC even in the absence of recurrent balancing selection. This effect is enhanced by high rates of recombination/gene conversion. The second hypothesis also assumes one specific form of selection, namely divergent selection. This is not necessarily the most obvious hypothesis to be made for MHC.

To me it therefore does not come as a surprise that neither of the predicted phylogenetic patterns is found. From all the phylogenies based on MHC IIB that I have come across, the actual result is indeed probably what I would have expected, although support of interspecific clusters might perhaps be somewhat more substantial here than elsewhere. In the discussion the authors then mention that “If either selection or divergence time were insufficient to shape MHC polymorphism in either a divergent or convergent manner, then we would expect the sequences among the species to reflect a neutral model[…]“. This is not correct, because divergent evolution and convergence are not the only and, importantly, not the primary modes of evolution of MHC genes. An expected pattern, which has been demonstrated in many studies (and is also strongly reflected in the results of the present paper to the extent that can be judged based on what is presented!) is that of trans-species polymorphism.

The authors state that “The mtDNA and nuclear intronic phylogenies show that sufficient time has passed since species divergence for neutral mutations to establish well supported clades […]. In my opinion there is actually no data to support this claim, simply because this conclusion cannot be made based on divergence time alone. The extent to which shared ancestral variation is sorted, i.e. the rates of lineage sorting depend both on effective population size (Ne) and the strength and direction of selection. The most important point here is that balancing selection can lead to the retention of ancestral variants across speciation events; we are back to trans-species polymorphisms. The authors discuss this possibility later on, stating that trans-species polymorphisms would be the results of ancient selection. In my opinion this is not entirely correct, because selection has to act recurrently (i.e. also in the present) in order to retain ancient variants (unless Ne is very large and ILS strong).

The authors make the point in the discussion, that overall the sequences cluster by species, while non-synonymous variation shows interspecific clusters. This would indeed indicate that the historical relationships are different from the ones observed for functional variation, and be (very interesting!) evidence for convergence. However, in my opinion there is not sufficient support for this result. Figure S4 shows a topology “of Synonymous Substitutions”. First, it is unclear what the authors mean by this; the rate of synonymous substitutions (dS), or sites with synonymous substitutions? Second, this topology contains a way lower number of sequences than Figure 3, and, importantly, also shows interspecific clusters (though they are not supported). Obtaining conclusive evidence for convergence for MHC IIB exon 2 sequences, which are strongly impacted by recombination, may be difficult, because bootstrap support will always be low. One possibility might be to compare the support for alternative topologies using SH-type tests for different data partitions.

Finally there are numerous problems with the writing. Many formulations are vague or imprecise, and the logic of a number of sentences is flawed. Also, English should be thoroughly revised.



Detailed comments:

Abstract
- How recently diverged are the study species?
- Which regions of MHC IIB has been studied?

Introduction
- L7-9: Evolutionary forces are mixed with molecular mechanisms. I think makes sense to make a better distinction between those.
- L24: I am not sure to agree that there is little evidence for contemporary selection on MHC in the wild.
- L36: What are “genetic patterns”?
- L38: Also demography can have an important effect.
- L39: “recombining loci” >> non-homologously recombining loci
- L42: “gene phylogeny” >> “gene history”
- L43-46: I do not understand this sentence.
- L48: What are “neutral molecular events”?
- The authors state that they want to differentiate between ancient and recent positive selection. The time scales which they consider ancient and recent are not necessarily clear, nor make necessarily much sense I am afraid. Surely, the carrion-American crow split is more recent, but would we really refer to everything happening after this as recent? Is 450,000 years recent, or do we refer to contemporary events with ‘recent’?
- L65-66: What are “contrasting phylogenetic relationships” and “contrasting geographic ranges”?
- L74/89: “In this analysis”, “this study” >> In the present analysis/study
- L91: “relatively recent” > see point above
- L95: “We analyzed data” > which kind of data?
- First hypothesis: Which type of selection would be acting here? Independent of this, how realistic is it to assume that selection acted only before, but not after the species split?

Materials and Methods
- L137: Where is this primer situated?
- L183: Which 454 platform was used for sequencing?
- L217: There is a recent study dealing with amplification biases which the authors may want to cite (Burri et al. 2014, Molecular Ecology Resources).
- L245: Cite the reference for the Bayesian approaches.
- L262: “MHC repertoires” >> MHC IIB repertoires
- L269 : ”Corvis ” >> Corvus
- L284 : The authors refer to net nucleotide divergence as dN. I think that this can be confusing, and given that it is usually referred to as dA could be avoided.

Results
- L313-315: Repetition of Methods, and could probably be left out.
- Jungle crow has more alleles than the other two species. Given that the primers were developed in Jungle crow, and MHC genes evolve rapidly, could this speak fr an allelic dropout due to mutations in the priming sites?
- L324: A variable number of variants detected between individuals does not necessarily indicate CNV. There are many instances of shared alleles among MHC paralogs.
- L341- : Why are the authors referring to the scrub jay here? Please cite relevant literature.


Discussion
- The discussion is lengthy, and should be shortened. I think this can easily be done by concentrating on the actual results, once these have been worked out clearly. E.g. focus on trans-species evolution rather than discussing the two original hypotheses (which might anyway be revised or completed with others in a revision).
- I disagree with the first sentence, as it disregards a number of other factors as outlined before.
- I think the conclusions should be completely revised, as I do not see much evidence for them as they currently stand. I am not convinced that the sharing of a single supertype among the sympatric species is evidence enough to claim convergence.

---

## Round 0.2 · Minor Revisions

Dear Dr. Eimes -

You might consider making the title of the paper more specific, for
example: Patterns of MHC evolution in MHC class II genes of crows
(Corvus); or even, Patterns of evolution of MHC class II genes of crows
(Corvus) suggest trans-species polymorphism. I think a more detailed title will serve the paper well.

Thank you for your extensive revision of your manuscript on MHC of crows. You seem to have responded to the reviewers' comments adequately. However, since the manuscript was so extensively re-written I felt it important to quickly read through it and make some comments myself. I have appended these comments to the end of this paragraph. Hopefully incorporating these suggestions will not take too much time. Thank you for your patience and consideration.

Abstract:

“Phylogenetic
trees constructed from putatively selected sites were better supported than those
constructed from putatively neutral sites, which suggests that recombination, especially
gene conversion, has partially erased the signal of allelic ancestry in these species.”

Please consider revising this sentence; you don’t indicate that the phylogenies are different, just that they are supported differently. The inference of recombination is therefore unclear.

2nd to last line of abstract: Perhaps include “overall” before “phylogenetic patterns”, to distinguish from previous phrase.

Line 44-45: May also want to cite this as more recent evidence for TSP: https://peerj.com/articles/86/ (in passerine class I genes). Possibly relevant for line 391 as well.

Line 74 and elsewhere - I don’t think the ‘mt’ of mtDNA needs to be italicized.

Consider removing the acronym “PSS” from the manuscript. It is only used about 4 times and doesn’t seem to warrant an acronym.

Line 284: Please provide units for the substitution rates indicated. Same for lines 367-373.

Lines 368-373: No need to italicize “Ma”. Sometimes the term “MY” is used to denote “millions of years”. I will leave it to the authors to decide what is most clear.

Line 377: remove “These”, since from this sentence alone it is not clear which crow species you are referring to.

Line 378: Perhaps insert “(Corvus sp.)” after crows. Some readers will skip straight to the discussion and they may not know what crow species you are referring to – or even what a crow is!

Figure 4. I don’t see any scale on the axes of the Eigenvalue graph or the discriminant function plot. Can this be corrected? At the very least the different Eigenvalues can be numbered 1-7. Plus the color of the bars is not clear (gray, black, white).

---

## Round 0.3 · accepted · Accept

Thank you for your revisions.